# Restoring Apoptosis with BH3 Mimetics in Mature B-Cell Malignancies

**DOI:** 10.3390/cells9030717

**Published:** 2020-03-14

**Authors:** Maxime Jullien, Patricia Gomez-Bougie, David Chiron, Cyrille Touzeau

**Affiliations:** 1Clinical Hematology, Nantes University Hospital, 1 place A. Ricordeau, 44000 Nantes, France; maxime.jullien@chu-nantes.fr; 2CRCINA, INSERM, CNRS, Angers University, Nantes University, 8 quai Moncousu, 44000 Nantes, France; patricia.gomez@inserm.fr (P.G.-B.); david.chiron@univ-nantes.fr (D.C.); 3Integrated Cancer Research Center (SIRIC), ILIAD, 5 Allée de l’Ile Gloriette, 44093 Nantes, France

**Keywords:** venetoclax, BH3 mimetics, BCL-2, MCL-1, apoptosis, myeloma, lymphoma, leukemia

## Abstract

Apoptosis is a highly conserved mechanism enabling the removal of unwanted cells. Mitochondrial apoptosis is governed by the B-cell lymphoma (BCL-2) family, including anti-apoptotic and pro-apoptotic proteins. Apoptosis evasion by dysregulation of anti-apoptotic BCL-2 members (BCL-2, MCL-1, BCL-X_L_) is a common hallmark in cancers. To divert this dysregulation into vulnerability, researchers have developed BH3 mimetics, which are small molecules that restore effective apoptosis in neoplastic cells by interfering with anti-apoptotic proteins. Among them, venetoclax is a potent and selective BCL-2 inhibitor, which has demonstrated the strongest clinical activity in mature B-cell malignancies, including chronic lymphoid leukemia, mantle-cell lymphoma, and multiple myeloma. Nevertheless, mechanisms of primary and acquired resistance have been recently described and several features such as cytogenetic abnormalities, BCL-2 family expression, and ex vivo drug testing have to be considered for predicting sensitivity to BH3 mimetics and helping in the identification of patients able to respond. The medical need to overcome resistance to BH3 mimetics supports the evaluation of innovative combination strategies. Novel agents including MCL-1 targeting BH3 mimetics are currently evaluated and may represent new therapeutic options in the field. The present review summarizes the current knowledge regarding venetoclax and other BH3 mimetics for the treatment of mature B-cell malignancies.

## 1. Introduction

Apoptosis is a cell death pathway by which a cell triggers its self-destruction in response to death signals. This physiological process is essential for the development and homeostasis of multicellular organisms, and its dysregulation is a common hallmark in cancer, increasing the survival of neoplastic cells and their resistance to treatment [1]. The apoptotic machinery is initiated by intracellular or extracellular death-inducing signals that are integrated via two distinct pathways, a mitochondrial mediated (intrinsic) and death receptor dependent (extrinsic) pathway. Following an apoptotic insult, apoptogenic factors, such as cytochrome c, are released from mitochondria and trigger the activation of caspases, which degrade cellular components and prepare their clearance with minimal stress to surrounding tissues [2].

B-cell lymphoma 2 (BCL-2) family proteins are key regulators of the mitochondrial mediated apoptosis and are structurally characterized by the presence of one of more BCL-2 Homology (BH) domain (BH1–BH4). In terms of function, BCL-2 family includes pro-apoptotic and anti-apoptotic members, as shown in Figure 1A.

BH3-only activators (BID, PUMA and BIM) directly bind and activate the pro-apoptotic pore-forming proteins BAX and BAK, whose activation and oligomerization form macropores in the mitochondrial outer membrane, resulting in the commitment of apoptosis via mitochondrial outer membrane permeabilization and the release of apoptogenic proteins leading to the activation of caspases. The multidomain anti-apoptotic proteins of BCL-2 family (BCL-2, BCL-X_L_, MCL-1, BCL-W, BFL-1) inhibit apoptosis by binding and sequestering the pro-apoptotic proteins, via the interaction with their BH3 domain. The BH3-only pro-apoptotic sensitizing proteins (BAD, BIK, NOXA, HRK, BMF) cannot directly activate BAX and BAK, but antagonize the action of anti-apoptotic proteins by releasing the BH3-only activators and/or the constitutive activated BAX/BAK through competitive binding. For apoptosis to occur, anti-apoptotic proteins within the cell must be overwhelmed by pro-apoptotic proteins [3].

Overexpression of the anti-apoptotic proteins of the BCL-2 family is an apoptosis resistant mechanism frequently observed in neoplasia, and especially in mature B-cell neoplasms, where anti-apoptotic genes are constantly upregulated compared to normal counterpart cells [4]. This survival mechanism is a major weak point, as the dissociation of these complexes by releasing pro-apoptotic proteins triggers the apoptosis of the tumor cells. This condition is known as “primed for death” and is the basis for the development of BH3 mimetics compounds that specifically target anti-apoptotic proteins of the BCL-2 family (Figure 1B) [5,6].

BH3-mimetics are small organic molecules mimicking BH3-only proteins, and they bind with high-affinity and specificity anti-apoptotic proteins of the BCL-2 family via their hydrophobic groove. Several molecules have paved the way for the development of BH3 mimetics. ABT-737 was the first compound targeting BCL-2, BCL-X_L_ and BCL-w, but has low solubility and oral bioavailability. ABT-263 (navitoclax) was another potent inhibitor of BCL-2, BCL-X_L_ and BCL-w, showing activity in chronic lymphocytic leukemia (CLL), but its clinical development was halted due to dose-limiting severe thrombocytopenia resulting from BCL-X_L_ inhibition [7]. The need to develop a BCL-2 inhibitor sparing BCL-X_L_ led to the development of venetoclax, and orally bioavailable selective inhibitor of the protein BCL-2, and the leading molecule of this new drug family [8].

## 2. Efficacy of Venetoclax in Mature B-Cell Neoplasms

Given its major role in mature B-cell neoplasms’ survival and resistance to therapy, important efforts have been made to target anti-apoptotic proteins of the BCL-2 family in the past years. Venetoclax is the first-in-class orally bioavailable BCL-2 selective BH3 mimetic that binds BCL-2 with high affinity (Ki < 0.01 nM) while sparing BCL-X_L_ (Ki = 48 nM) and MCL-1 (Ki = 444 nM) [8].

### 2.1. Clinical Activity of Venetoclax in Chronic Lymphocytic Leukemia

BCL-2 is overexpressed in CLL [4] and BH3 profiling analysis demonstrated that circulating cells are mostly dependent on BCL-2 for survival [9]. Several clinical studies therefore evaluated venetoclax as a single agent or in combination in CLL patients (Table 1).

The first evidence of strong clinical activity of venetoclax single agent at the dose of 400 mg/day was observed after a ramp-up was obtained in the relapsed/refractory (RR) setting in the M12-175 phase I dose-escalation study [10]. Interestingly, reflecting pre-clinical data showing a TP53-independent induced apoptosis of CLL cells [11], venetoclax then showed a potent activity in high-risk CLL harboring a 17p deletion in a phase II study [12], leading to US Food and Drug Agency (FDA) approval in this setting in 2016. The updated results of this study [13] showed an overall response rate (ORR) of 77% and minimal residual disease (MRD) negativity of 30% with a sensitivity of 10^−4^, in a population of CLL patients with 17p deletion, mostly in the relapse setting.

Still in the RR setting, venetoclax in association with anti-CD20 monoclonal antibody rituximab proved its superiority over bendamustine in the phase III MURANO study, with a benefit in terms of progression-free survival (PFS): median not reached vs 17 months, and a hazard ratio (HR) for progression or death of 0.17 (*p* < 0.001) [14]. Interestingly, the benefit of venetoclax persisted even in the high-risk setting (presence of a 17p deletion, TP53 mutation, or unmutated IgHV genes). This study led to the FDA and European Medicines Agency’s (EMA) approval of venetoclax in combination with rituximab for the treatment of previously treated CLL in 2018.

In the frontline setting, venetoclax in association with anti-CD20 antibody obinutuzumab demonstrated its superiority over chlorambucil obinutuzumab in patients with CLL and coexisting conditions (score greater than 6 on the Cumulative Illness Rating Scale or a calculated creatinine clearance of less than 70 mL/min) [15], with a HR for progression or death of 0.35 in favor of venetoclax (*p* < 0.001), and a 24 months PFS of 88% vs. 64%. Again, this benefit was also observed in high risk patients. This study led to the FDA approval of venetoclax in combination with obinutuzumab for the treatment of untreated CLL in 2019.

Recently, venetoclax demonstrated impressive clinical activity in combination with BTK inhibitor ibrutinib in a phase II study involving previously untreated high-risk and older patients with CLL [16]: 92% of the patients had unmutated IgHV, TP53 aberration, or chromosome 11q deletion. The complete response rate after 12 cycles of combined treatment was 88%, and 61% of patients underwent remission with an undetectable MRD with sensitivity of 10^−4^. These results support ex vivo dynamic BH3 profiling data suggesting that BTK inhibition enhances mitochondrial BCL-2 dependence [17].

### 2.2. Clinical Activity of Venetoclax in Non-Hodgkin Lymphoma

The results of the non-Hodgkin lymphoma (NHL) cohort of the phase I M12-175 study demonstrated significant albeit variable venetoclax single agent activity among NHL subtypes [18]. The highest response rate was seen in relapsed/refractory mantle-cell lymphoma (MCL), with an overall response rate (ORR) of 21/28 patients (75%) and 6 patients (21%) achieving complete response (CR). This high response rate is consistent with the fact that MCL cells are commonly found to overexpress BCL-2 [4,19,20]. Clinical activity was also observed among others NHL subtypes: ORR and CR rates being respectively of 38% and 14% in follicular lymphoma (FL) and 18% and 12% in diffuse large B-cell lymphoma (DLBCL).

A recent phase II study confirmed the strong clinical activity of venetoclax in combination with ibrutinib for the treatment of MCL, with a PET-assessed complete response rate of 71% in a small cohort (*n* = 24) of high-risk MCL patients (75% MIPI high risk) [21].

The recent phase Ib study CAVALI showed the favorable safety profile of venetoclax in combination with rituximab or obinutuzumab and cyclophosphamide, doxorubicin, vincristine, and prednisone (R-/G-CHOP) chemotherapy in NHL [22]. In this trial, venetoclax was given a shorter dosing schedule (5 days in cycle 1 and from day 1 to day 10 in cycles 2–8) in order to mitigate the risk of cytopenia. The efficacy of this combination is being evaluated in newly diagnosed DLBCL in the phase II portion of the study.

Other trials are currently evaluating venetoclax in combination for the treatment of NHL and are summarized in Table 2.

### 2.3. Clinical Activity of Venetoclax in Multiple Myeloma

In multiple myeloma (MM), preclinical data revealed that a high sensitivity to venetoclax (nanomolar range) was mostly observed in cell lines and primary MM cells harboring the t(11;14) translocation [23].

The clinical activity of single agent venetoclax in relapsed/refractory MM was confirmed in a phase I clinical trial [24]. In the dose escalation part of the study (*n* = 30), patients received venetoclax orally from 300 to 1200 mg/day until progression. In the safety expansion part of the study (*n* = 36), patients received venetoclax 1200 mg daily until progression, as no maximum tolerated dose was reached in the dose escalation part. Patients enrolled had very advanced MM with a median number of five prior therapies, and most were refractory to both bortezomib and lenalidomide. With the exception of 2 patients, all patients who achieved response were positive for the t(11;14) translocation. From these patients, the overall response rate (ORR) was 40% (including 14% CR and 13% very good partial response (VGPR)) and the median progression-free survival was 6.6 months (3.9–10.2).

Interestingly, in vitro studies in myeloma cell lines demonstrated the synergistic effect of venetoclax in combination with dexamethasone, via the increased expression of BIM and BCL-2, promoting BCL-2 dependence [25]. In combination with bortezomib and dexamethasone, venetoclax demonstrated a promising efficacy in relapsed MM in a phase I study [26]. In patients non-refractory to bortezomib, the ORR was 90%, including 28% CR and 36% VGPR. Based on these promising efficacy results, the combination of venetoclax bortezomib dexamethasone was compared to bortezomib dexamethasone in the phase III BELLINI study (NCT02755597). In patients with RR MM who received 1 to 3 prior lines of therapy, venetoclax bortezomib dexamethasone resulted in a significant PFS benefit (22.9 months vs 11.4 months, *p* = 0.001, HR 0.59) [27]. However, the venetoclax arm of the study was associated with an excess risk of treatment emergent deaths, mostly attributed to infectious causes. Interestingly, the trial confirmed excellent efficacy in patients harboring t(11;14) with an ORR of 90% including 46% CR. Notably, higher response rates and PFS were also observed in patients with high BCL-2 expression determined by immunochemistry.

Based on these results, the clinical development of venetoclax in MM is currently restricted to patients with t(11;14). Clinical trials currently evaluating venetoclax for the treatment of MM are indicated in Table 3.

## 3. Safety of BH3 Mimetics

### 3.1. Tumor Lysis Syndrome

The most important toxic effect reported in the dose escalating cohort of the phase I first-in-human study of venetoclax monotherapy in CLL was tumor lysis syndrome (TLS), occurring in 10 of 56 patients. The occurrence was clinical in three patients and resulted in the death of one [10]. The occurrence of TLS in the first three patients treated led to the introduction of a weekly dose ramp-up starting at the dose of 20 mg/day, prophylaxis of TLS by allopurinol and hydration, and hospitalization for the administrations of the first doses, and for the subsequent dose ramp-ups for patients considered to be at high risk for TLS. This strategy efficiently reduced the incidence and severity of TLS, as only one patient out of the 60 in the expansion cohort presented laboratory evidence of TLS, without clinical sequelae.

Clinical TLS was rarely observed in the NHL population [18,22] except in MCL, where it occurred in two of the first 15 patients treated with ibrutinib venetoclax combination [21]. In this study, venetoclax was started after four weeks of ibrutinib monotherapy. Notably, the two patients presenting TLS had not achieved a clinical response after four weeks of ibrutinib, and had started venetoclax at the dose of 50 mg daily. No cases of TLS were reported after lowering the starting dose of venetoclax to 20 mg daily.

No events of TLS were reported in patients treated for a MM [24,26].

### 3.2. Other Clinically Relevant Adverse Events

The most frequent grade 3–4 hematological adverse events reported with venetoclax monotherapy were neutropenia (11–41%), thrombocytopenia (12–26%), and anemia (12–15%). Venetoclax monotherapy was otherwise well tolerated, and the most frequent non hematological adverse event consisted of gastrointestinal side effects such as nausea (47–48%, including 2–3% of grade 3–4) and diarrhea (36–52%, including 2–3% of grade 3–4) [10,18,24], and an increased risk of upper respiratory tract infections (48%, including 1% grade 3–4) [10].

This favorable safety profile may be explained by preclinical data. In normal hematopoietic tissue, BCL-2 expression is mostly found in bone marrow precursor cells, in the mantle zone of germinal centers, as well as in the surviving T cells in the thymic medulla [28]. Interestingly, BCL-2 knock-out experiments in mice does not impair early hematopoiesis, but in those mice, mature lymphocytes undergo massive apoptosis within the first few weeks of life [29]. In non-hematopoietic tissues, BCL-2 expression is topically restricted to long-lived or proliferating cell zones, as in the epithelial regenerative compartment or the basal crypts of the normal colon and small intestine [28].

Notably, in combination with bortezomib and dexamethasone in the BELLINI phase III study, the interim analysis highlighted a higher risk of death in the venetoclax arm with the increase of treatment-emergent deaths due to infection, despite a significant augmentation of PFS [27]. Nevertheless, a clinical benefit was observed in the patients with t(11;14) or high expression levels of BCL-2, suggesting the interest of a biomarker-driven approach.

## 4. Resistance Mechanisms to Venetoclax

Although targeted therapies provided a recent breakthrough in cancer treatment, tumor cells are heterogeneous between patients and even within the tumor, and show a great ability to escape treatment. Venetoclax does not escape this logic and several mechanisms of primary or acquired resistance have recently been described in mature B-cell malignancies.

### 4.1. Mechanisms of Primary Resistance

Sensitivity to venetoclax does not only depend on the expression of BCL-2 but also on other members of the BCL-2 family, in particular MCL-1 and BCL-X_L_, which can partially compensate for the loss of BCL-2 function. Thus, the heterogeneous expression and imbalance of the BCL-2 family observed in malignant mature B-cell tumors often results in primary resistance to venetoclax [4]. Several studies have accordingly demonstrated a correlation between resistance to venetoclax and the relative expression of BCL-2 to MCL-1, BCL-X_L_ or BIM in MM, MCL or FL [23,30,31].

Among the molecular mechanisms leading to the imbalance of the BCL-2 family, genomic analyses of patients treated with ibrutinib and venetoclax in the AIM clinical cohort have identified mutations in the SWI-SNF chromatin remodeling complex, conferring resistance to treatment in MCL. Interestingly, the authors clearly demonstrated that abnormalities in SWI-SNF, through the loss of SMARCA2 (del9p) or SMARCA4 mutations, led to downregulation of ATF3, due to the loss of accessibility of the chromatin at its locus, and therefore overexpression of BCL-X_L_, ATF3 being a direct BCL-X_L_ repressor [32].

### 4.2. Mechanism of Acquired Intrinsic Resistance

The first publications focusing on intrinsic resistance to venetoclax were based on long-term treated cell lines to study the emergence of resistance. As might be expected, the resistant clones were characterized by specific profiles of the BCL-2 family, including an overall downregulation of the BH3-only proteins and/or upregulation of the antiapoptotic proteins MCL-1 and BCL-X_L_ [31,33,34]. In addition to the modulation of the expression of the BCL-2 family, these preclinical studies have described the acquisition of BCL-2 or BAX mutations, both leading to high resistance to venetoclax [33,34,35].

Although acquired BAX mutations have not been described so far in venetoclax resistant patients, clinical studies have recently detected BCL-2 mutations. Indeed, in CLL, two teams have reported the acquisition of point mutations in BCL-2 by analyzing paired pre-venetoclax and progression samples from patients enrolled on venetoclax clinical trials. Glycine substitution by Valin in position 101 (G101V) was firstly identified at progression in 7/15 patients (but not at study entry) [36]. In vitro experiments on CLL cell lines demonstrated that the cells harboring G101V were approximately 30-fold less sensitive to venetoclax, due to a 180-fold reduction of venetoclax binding. This mutation was independently reported by another team, along a second one (D103Y) [37]. Additional BCL2 mutations acquired in parallel with BCL-2 G101V were recently reported in patients with progressive CLL on venetoclax [38]. Acquired BCL-2 mutations have also been described in MCL, but recent studies suggested that they are infrequent [39]. In addition to mutations, copy number variation has also been involved in acquired resistance to venetoclax. Zhao et al. identified the selection of rare MCL cells in vitro having BCL-2 amplicon loss (18q) during venetoclax treatment [40]. In addition, an acquired amplification of the 1q23 region, encompassing MCL-1, has recently been described in CLL lines and patients progressing on venetoclax [41]. Since 1q23 amplification is also common in MM, one could predict that it might also be involved in venetoclax resistance [42].

In addition to the acquired resistance directly linked to changes in the BCL-2 family, Herling et al. have recently reported other potential mechanisms through whole-exome sequencing of serial samples from CLL patients treated with venetoclax [43]. Indeed, they identified recurrent CDKN2A/B deletions as well as BTG1 and BRAF mutations at relapse. While these results need to be confirmed in a larger cohort, they support the ability of B-cell malignancies to escape the pressure of targeted therapies in many ways.

### 4.3. Mechanism of Microenvironment-Dependent Resistance

The major role of tumor niches is now widely accepted in cancer, including malignant mature B-cell tumors [44,45]. Several groups have demonstrated that various factors present in the microenvironment, such as CD40L, cytokines, or TLR ligands, strongly modulate the expression of the BCL-2 family in mature B-cell malignancies [30,46,47,48].

. Even if the parameters of these studies are heterogeneous with regard to extrinsic models and stimuli, they all demonstrated positive regulation dependent on the microenvironment of at least one anti-apoptotic protein among BCL-X_L_, MCL-1, and BFL-1. This imbalance, also exacerbated by the simultaneous decrease in the BH3-only proteins, has led to loss of mitochondrial priming and therefore to resistance of venetoclax in vitro [49]. Together, these preclinical studies show that although circulating lymphoma cells appear to be primarily dependent on BCL-2, conferring high sensitivity to venetoclax, extrinsic signals can modulate this dependence. As a result, tumor cells in their protective niches may be more resistant to venetoclax than their circulating counterparts, highlighting the need to integrate the microenvironment into in vitro studies.

This microenvironment-dependent modulation of the BCL-2 family has also been confirmed in vivo in several B-cell malignancies thanks to the analysis of Affimetrix datasets, gathering samples from the peripheral blood and lymphoid organs [4]. Even if the modulation profile seems specific to each histology, the anti-apoptotic protein BCL-X_L_ was higher in the tissue compared to peripheral blood, regardless on the nature of malignant B-cells. In parallel, the pro-apoptotic BH3-only BIM and NOXA were deeply suppressed in all malignancies studied (CLL, MCL, FL, and SMZL) suggesting a central role of the tumor ecosystem in their regulation.

On the basis of these preclinical results, several groups have focused on the development of rational, mechanism-based combinations, which integrate the key role of the microenvironment. Oppermann and colleagues have developed a kinase inhibitor screen to identify small molecules capable of countering microenvironment-dependent resistance to venetoclax in CLL cells. Using this strategy, they showed that sunitinib, a multitargeted tyrosine kinase inhibitor, can overcome microenvironment-dependent resistance to venetoclax by simultaneously decreasing the expression of MCL-1, BFL-1, and BCL-X_L_ [48]. Similarly, another group has demonstrated that Cerdulatinib, a double SYK/JAK inhibitor, overcomes microenvironment resistance in vitro (CD40L, Cytokines, Nurse-like cells) and synergizes with venetoclax by inhibiting BCL-X_L_ and MCL-1 [50].

Among kinase inhibitors, the development of BTK inhibitors such as ibrutinib has been a breakthrough in the treatment of CLL and MCL. In addition to inhibiting BCR, the inhibition of BTK leads to an alteration in chemokines signaling, leading to an efficient egression of tumor cells from their protective niches [51,52]. This lymphocytosis provided strong biological justification for sequential treatment with venetoclax. Indeed, while tumor cells are predicted to be more resistant within their microenvironment, they are highly dependent on BCL-2 in peripheral blood. This hypothesis was confirmed in CLL and MCL in vitro and supported the rationality of the sequential association ibrutinib/venetoclax in vivo [17,49]. Likewise, preclinical studies have also supported the addition of CD20 antibodies (rituximab, obinutuzumab) to venetoclax in vivo. Indeed, it has been shown that the type II anti-CD20 antibody, obinutuzumab, and to a lesser extent the standard of care rituximab, counteracted the microenvironment-dependent BCL-X_L_ induction by inhibiting NF-kB in primary CLL and MCL cells [47,49]. Observed together, these ex vivo data supported the rationality of the tri-targeted therapy ibrutinib, obinutuzumab, and venetoclax in CLL and MCL (Table 2).

### 4.4. Venetoclax Resistance due to Mitochondria Reprogramming and Modulation of Intermediate Metabolism

The most common recurrent mechanism of venetoclax resistance in patients with CLL is the acquisition of Gly101Val BCL-2 mutation, impairing the venetoclax binding. Nevertheless, this mutation was found at subclonal levels and only in a subset of patients [36], suggesting the presence of other resistance mechanisms in these patients. Recent studies improve the current understanding of how targeting energy use and mitochondrial homeostasis may lead to enhancement of venetoclax response. Indeed, Guièze et al. reported the involvement of mitochondrial reprogramming leading to increase oxidative phosphorylation (OXPHOS) in venetoclax resistant CLL cells, which was accompanied with higher steady state levels of reactive oxygen species and higher mitochondrial membrane potential in resistant cells compared to parental cell lines [41]. These finding were explained by an increased mass of mitochondria per cell and demonstrated that venetoclax contributes to a broader scope of “mitochondrial stress”, beyond the expression/mutations of BCL-2 members.

However, other energy pathways could also be involved in venetoclax resistance [41]. Accordingly, targeting intermediary metabolism can be a novel strategy to enhance the efficacy of BH3 mimetics in hematologic malignancies. Thus, targeting glutamine uptake and its downstream signaling pathway, such as glutaminolysis, reductive carboxylation, lipogenesis, and cholesterogenesis, induced a marked sensitization of resistant cells to BH3 mimetics in CLL, MCL, and MM models [53]. In agreement, another preclinical study demonstrated that the combination of statins with venetoclax was cytotoxic to DLBCL, CLL, and AML cells and reduced lymphoma burden in a syngeneic mouse model of BCL-2/MYC driven “double hit” lymphoma [54]. Consequently, the retrospective study of CLL patients found that those who were statins users presented an enhanced venetoclax clinical response, demonstrated by more frequent complete responses. These last findings highlight the possibility of proposing widely used treatments, as statins, to target intermediary metabolism and in this way ameliorate the efficacy of BH3 mimetics therapy.

## 5. Biomarkers

While most circulating CLL and MCL cells are characterized by their dependence to BCL-2 for survival, MM and other mature B-cell malignancies are heterogeneous diseases regarding anti-apoptotic protein dependence for survival. The development of biomarkers able to identify patients who may benefit from venetoclax therapy is therefore essential. In that perspective, several strategies have been developed.

### 5.1. Cytogenetic Abnormalities

In multiple myeloma, the presence of t(11;14) translocation is strongly associated with sensitivity to venetoclax single agent in vitro [23]. The phase 1–3 clinical trials have confirmed the predictive value of the t(11;14) translocation to predict clinical response to venetoclax as a single agent or in combination [24,27]. No biomarkers based on cytogenetic has been shown to predict response to venetoclax in other mature B-cell malignancies.

### 5.2. BCL-2 Family Member Expression

In MM, in vitro sensitivity to venetoclax from cell lines and primary samples has been analyzed in accordance to the expression of the anti-apoptotic proteins BCL-2, BCL-X_L_, and MCL-1 [23,55]. In those two studies, sensitivity to venetoclax was associated with a higher expression of BCL-2 relative to MCL-1 and BCL-X_L_. In the phase 1 trial evaluation of venetoclax monotherapy in relapsed/refractory MM patients, BCL-2, BCL-X_L_, and MCL-1 gene expression was quantified by droplet digital polymerase chain reaction (PCR) (*n* = 44). The ratios of BCL-2/MCL-1 and BCL-2/BCL-X_L_ were significantly higher in patients who achieved an overall response to venetoclax [24]. Interestingly, a high ratio of BCL-2/BCL-X_L_ was more frequent in the t(11;14) than in the non-t(11;14) subgroup (38% vs 5% respectively). Moreover, BCL-2/BCL-X_L_ ratio appeared to identify responders more efficiently than t(11;14), as in the t(11;14) subgroup, patients with a low BCL-2/BCL-X_L_ ratio (*n* = 15) had a lower ORR than patients with a high BCL-2/BCL-X_L_ ratio (*n* = 9) (20% vs 88% respectively). In addition to MM, a specific expression ratio could be effective in predicting the sensitivity of cells from other mature B-cell malignancies. Therefore, based on the expression of the previously described factors involved in venetoclax resistance (MCL-1, BCL2L1, BFL-1) and factors known to be involved in the effectiveness of venetoclax (BCL-2, BCL2L11, BAX), we have determined that the ratio (BCL-2 + BCL2L11 + BAX)/BCL2L1 was the most powerful predictor of the response to venetoclax for mature B-cell malignancies [4].

### 5.3. BH3 Mimetics Ex Vivo Testing

A major challenge for the therapeutic application of BH3-mimetics is to match the right targeted treatment to the right patient. Indeed, one of the major issues for the use of BH3 mimetics is the lack of specific biomarkers to guide their use in the clinic. One approach to tackle this issue in patient samples is the ex vivo challenge of primary cells with BH3-mimetics. The read out of this ex vivo testing is cell death, evaluated by flow cytometry. In this approach, the extent of induced cell death directly reflects the dependence to a given anti-apoptotic protein. The ex vivo testing was used to demonstrate the cell dependence to BCL-2 anti-apoptotic proteins in human B-cell tonsils and CLL primary cells, as well as primary cell from MM and DLBCL [56,57,58]. This functional approach is valuable not only to identify dependence to a single anti-apoptotic member but also to determine co-dependencies, as observed in MM [57]. Finally, the clinical utility of this functional assay was confirmed in a recent study demonstrating that the ex vivo sensitivity to venetoclax of primary MM cells predicts the clinical response in myeloma [59]. Thus, the above-mentioned studies are the proof of concept that the BH3 mimetics ex vivo testing is one of the most suitable strategies to guide the clinical use of these novel compounds.

### 5.4. BH3 Profiling

BH3 profiling is a functional assay that identifies the dependence of live cancer cells to anti-apoptotic members of BCL-2 family and defines the “apoptotic priming”, i.e., the proximity of a cell to the apoptotic threshold [60]. In this assay, cells are exposed to peptides derived from the BH3 domains of proapoptotic BH3-only proteins of the BCL-2 family, the resulting mitochondrial outer membrane permeabilization is indirectly measured by flow cytometry analysis.

In MM, pretreatment mitochondrial priming measured by BH3 profiling technique correlates with clinical response to cytotoxic chemotherapy [6]. BH3 profiling also allows for the identification of BCL-2 dependence in myeloma cells and to predict the response to venetoclax [61]. In CLL cells, BH3 profiling revealed that CLL cells from the peripheral blood are highly primed, and that this increased priming is associated with improved clinical response [62]. On the contrary, decreased mitochondrial apoptotic priming of CLL cells is associated with worse clinical response to chemotherapy.

Notably, dynamic BH3 profiling, evaluating early changes to apoptosis signaling following drug perturbation in vitro, was shown to predict cytotoxic response of various cancers to chemotherapeutics in vivo [63].

## 6. BH3 Mimetics Targeting MCL-1

As previously mentioned, high levels of MCL-1 and BCL-X_L_ are major mechanisms of resistance to venetoclax in mature B-cell malignancies. There is currently no BCL-X_L_ selective inhibitor for clinical use, but the efficacy of several specific MCL-1 inhibitors is currently being tested in humans. The antiapoptotic protein MCL-1 is an irrefutable therapeutic target in hematologic malignancies. Its role as a critical survival factor for many tumor types, including MM and MCL, is well documented [64,65]. Outstanding progress has been recently made in the discovery of selective and potent MCL-1 BH3 mimetics and several compounds are currently investigated in phase 1 clinical trials in mature B-cell malignancies (Table 4). Despite the particular chemical features of each compound, they all exhibit a high specificity for MCL-1 and were reported to induce apoptosis in MM, MCL, and CLL pre-clinical studies [66,67,68]. It is notable that the addiction to MCL-1 of primary cells from MM patients at relapse is increased compared to those at diagnosis [57]. Furthermore, preclinical studies have demonstrated the rationale to combine venetoclax and BH3 mimetic targeting MCL-1 in MCL and MM [69,70].

An important aspect to take into account when considering MCL-1 BH3 mimetics for clinical applications is the key pro-survival role of MCL-1 in many normal tissues, particularly cardiomyocytes. As these conclusions were drawn mostly from conditional MCL-1 knock-out mice, it is possible that the observed effects in normal tissues are caused by chronic depletion of MCL-1 or the inhibition of other functions besides its anti-apoptotic role [71,72]. Therefore, a careful dosing and schedule is needed to potentially mitigate the side effects of MCL-1 BH3 mimetics. Indeed, the scientific community expects that the current clinical evaluation of these MCL-1 BH3 mimetics in hematologic malignancies will result in at least one of them being clinically approved.

## 7. Conclusions

BH3 mimetics are an innovative class in the therapeutic arsenal of onco-hematology. Venetoclax is a first-in-class, orally available molecule with an acceptable risk profile. Its efficacy has been demonstrated in particular in CLL, even in high cytogenetic risk subgroups, leading to several FDA approvals, and in MCL. In MM, sensitivity to venetoclax appears to be restricted to the subgroup of patients with t(11;14) translocation and/or high BCL-2 expression. In this context, the development of biomarkers allowing for the prediction of the clinical response seems essential in order to select the patients who benefit the most from such therapies more efficiently. New BH3 mimetics, including MCL-1 targeted compounds, are currently under clinical trials and might provide valuable new options for patients with hematologic malignancies. Still, the mechanisms of resistance to these new molecules are varied, and it is very likely that the future of BH3 mimetics will lie in combinations, whether with conventional chemotherapies or targeted therapies such as BTK inhibitors, or in combinations of different BH3 mimetics.

## Figures and Tables

**Figure 1 cells-09-00717-f001:**
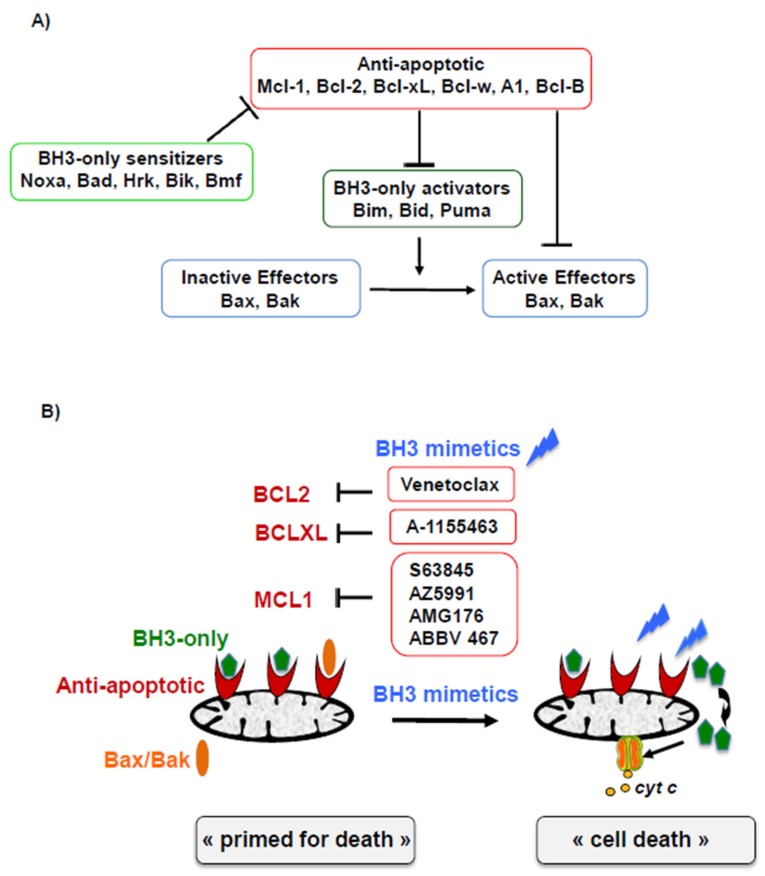
Interconnected network of B-cell lymphoma (BCL-2) family proteins. (**A**) The BCL-2 family consists of three groups: anti-apoptotic members (MCL-1, BCL-2, BCL-X_L_, etc.), pro-apoptotic BH3-only proteins, and the pro-apoptotic effectors BAX and BAK. BH3-only proteins can activate BAX-BAK either directly (BH3-only activators), inducing conformational changes of BAX/BAK or indirectly (BH3-only sensitizers), interacting with the anti-apoptotic counterparts and in turn inducing the release of constitutive activated effectors BAX/BAK. Cancer cell constrain pro-apoptotic proteins via physical interactions with anti-apoptotic counterparts, termed “primed for death”. (**B**) BH3 mimetics bind specifically anti-apoptotic BCL-2 members, releasing pro-apoptotic BH3-only and/or effectors BAX/BAK, which in turn trigger the apoptotic cascade leading to cell death.

**Table 1 cells-09-00717-t001:** Results from clinical trials evaluating venetoclax single agent or in combination in CLL patients.

Reference	Phase	Drugs	Population	Efficacy	Survival
Roberts 2016 [10]	I	Venetoclax single agent	116 Patients with RR CLL	ORR 79 %, CR 20%	15m PFS 66%
Stilgenbauer 2016 [12] and 2018 [13]	II	Venetoclax single agent	158 Patients with del17p CLL	ORR 77%, MRD- 30%	24m PFS 54%
Seymour 2018 [14]	III	Venetoclax Rituximab	389 Patients with RR CLL	MRD- 83.5%	24m PFS 85%
Fisher 2019 [15]	III	Venetoclax Obinutuzumab	432 Patients with untreated CLL and comorbidities ^α^	MRD- 75.5%	24m PFS 88%
Jain 2019 [16]	II	Venetoclax Ibrutinib	80 Patients with untreated CLL, and high-risk criteria (92%) and/or old age (30% > 70y)	CR 88%, MRD- 61%	12m PFS 98%

Legend: High risk criteria: del17p, del11q, TP53 mutation or unmutated IGHV. ORR: overall response rate. CR: complete response. MRD: minimal residual disease, sensibility 10^−4^. PFS: progression-free survival. RR: relapsed/refractory. CLL: chronic lymphoid leukemia ^α^: comorbidities: calculated creatinine clearance < 70mL/min or Cumulative Illness Rating Scale > 6.

**Table 2 cells-09-00717-t002:** Selected ongoing clinical trials evaluating venetoclax in combination for the treatment of mature B-cell malignancies excluding multiple myeloma (MM).

Venetoclax +	Phase	Disease	Clinical Trials Identifier
ibrutinib obinutuzumab	I/II	CLL	NCT02427451
ibrutinib obinutuzumab	II	CLL	NCT03755947
acalabrutinib obinutuzumab	II	frontline CLL	NCT03580928
obinutuzumab bendamustine	II	frontline FL	NCT03113422
lenalidomide rituximab	I/II	RR MCL	NCT03505944 (VALERIA)
ibrutinib obinutuzumab	I/II	RR and frontline MCL	NCT02558816 (OAsIs)
bendamustine rituximab	II	frontline MCL	NCT03834688
bendamustine obinutuzumab	II	frontline MCL	NCT03872180
ibrutinib (compared to placebo + ibrutinib)	III	RR MCL	NCT03112174 (SYMPATICO)

Legend: RR: relapsed/refractory. FL: follicular lymphoma. CLL: chronic lymphocytic leukemia. MCL: mantle cell lymphoma. MM: multiple myeloma. Pts: patients.

**Table 3 cells-09-00717-t003:** Selected ongoing clinical trials evaluating venetoclax in combination for the treatment of MM.

Venetoclax +	Phase	Disease	Clinical Trials Identifier
carfilzomib dexamethasone	II	RR MM	NCT02899052
daratumumab dexamethasone	II	RR MM with t(11;14)	NCT03314181 part 1–3
daratumumab bortezomib dexamethasone	II	RR MM with t(11;14)	NCT03314181 part 2
dexamethasone (compared to pomalidomide dexamethasone)	III	RR MM with t(11;14)	NCT03539744 (CANOVA)

Legend. RR: relapsed/refractory.

**Table 4 cells-09-00717-t004:** New BH3 mimetics under clinical development.

	Administration	Phase	Disease	Clinical Trials Identifier
**MCL-1 inhibitors**				
AMG 176	IV	I	RRMM	NCT02675452
AMG 397	PO	I	RRMM, RR NHL	NCT03465540
AZD-5991	IV	I	RRMM	NCT03218683
ABBV-467	IV	I	RRMM	NCT04178902
MIK665	IV	I	RRMM, RR NHL	NCT02992483
**BCL-2 inhibitors**				
S65487	IV	I	RRMM, RRNHL	NCT03755154
APG-2575	PO	I	RRNHL, RR CLL	NCT03913949

IV: intra-venous administration. PO: oral administration. RR: relapsed/refractory. MM: multiple myeloma. NHL: non-Hodgkin lymphoma. Pts: patients.

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
