# Peer review of "Restoring Apoptosis with BH3 Mimetics in Mature B-Cell Malignancies"

_cells, 2020, doi:10.3390/cells9030717_

Round 1

Reviewer 1 Report

Overall this a well-written manuscript and covers a broad overview of what is a burgeoning field of the use of apoptosis-inducing drugs in onco-hematology.

A number of points:

Figure 1: Ensure consistency between the figure and text. For example, Figure 1 references A1, text usually refers to as BFL-1 though in some places BCL2A1. In addition, ensure either BCL-XL or BCL-XL. In all cases, are the family names hyphenated or not. Figure 1: Should perhaps indicate other well-known BH3-mimetics including ABT-263 (BCL-XL, BCL-W, BCL-2), A1331852 (BCL-XL), S64315 (MCL-1) for completeness. As this is a review, I think in a number of places a more comprehensive reference list should be included, especially where more recent papers have been published. For example, for the concept of “primed for death” the authors only referenced Certo et al. Multiple other high-profile papers describe this phenomenon, for e.g. Ni Chonghaile T Science 2011. As another example, the authors mention that venetoclax has shown potent activity in several mature B cell neoplasms, yet only one reference to Souers et al was made. Table 2: Appears incorrect. Should this be for trials for the treatment of NHL? Section 3.2: In addition to the other clinically relevant adverse events described, should also include increased risk of respiratory tract infections (Roberts AW NEJM 2016). Section 4.2: Should include reference to Blombery P Blood 2020 describing identification of multiple BCL-2 mutations accompanying the originally identified G101V. Section 5.4: Expand section to describe the progress made with BH3-profiling. For example, the idea of dynamic BH3 profiling where changes to apoptosis signalling is interrogated following drug perturbation (Montero J Cell 2015).

Reviewer 2 Report

The manuscript written by Jullien M and colleagues is about the use of small organic compounds called BH3 mimetics in the treatment of B-cell malignancies.

The manuscript majorly articulates around venetoclax a molecule that targets the Bcl-2 protein. The text is well written and there is extensive literature research concerning the state of the art of the past and current trials on venetoclax.

Although the introduction section is quite important it would be interesting for the reader to have a brief historic overview about the discovery of the first BH3 mimetic molecules.

In this context it worth nothing to mention the reasons to develop ABT-199 in regards to the side effects of ABT-263 towards platelets.

Authors may discuss a little bit more the possible side effects of venetoclax on non-apoptotic Bcl-2-dependent functions as this has been briefly mentioned for Mcl-1 (line 397).

The figure 1B needs improvement as it is a bit squeezed.
